# Foliar Spraying of *Solanum tuberosum* L. with CaCl_2_ and Ca(NO_3_)_2_: Interactions with Nutrients Accumulation in Tubers

**DOI:** 10.3390/plants11131725

**Published:** 2022-06-29

**Authors:** Ana Rita F. Coelho, José Cochicho Ramalho, Fernando Cebola Lidon, Ana Coelho Marques, Diana Daccak, Cláudia Campos Pessoa, Inês Carmo Luís, Mauro Guerra, Roberta G. Leitão, José Manuel N. Semedo, Maria Manuela Silva, Isabel P. Pais, Nuno Leal, Carlos Galhano, Ana Paula Rodrigues, Paulo Legoinha, Maria José Silva, Maria Simões, Paula Scotti Campos, Maria Fernanda Pessoa, Fernando Henrique Reboredo

**Affiliations:** 1Departamento Ciências da Terra, Faculdade de Ciências e Tecnologia, Universidade NOVA de Lisboa, Campus da Caparica, 2829-516 Caparica, Portugal; fjl@fct.unl.pt (F.C.L.); amc.marques@campus.fct.unl.pt (A.C.M.); d.daccak@campus.fct.unl.pt (D.D.); c.pessoa@campus.fct.unl.pt (C.C.P.); idc.rodrigues@campus.fct.unl.pt (I.C.L.); n.leal@fct.unl.pt (N.L.); acag@fct.unl.pt (C.G.); pal@fct.unl.pt (P.L.); mmsr@fct.unl.pt (M.S.); mfgp@fct.unl.pt (M.F.P.); fhr@fct.unl.pt (F.H.R.); 2Unidade de Geobiociências, Geoengenharias e Geotecnologias (GeoBioTec), Faculdade de Ciências e Tecnologia, Universidade NOVA de Lisboa, Campus da Caparica, 2829-516 Caparica, Portugal; cochichor@mail.telepac.pt (J.C.R.); jose.semedo@iniav.pt (J.M.N.S.); abreusilva.manuela@gmail.com (M.M.S.); isabel.pais@iniav.pt (I.P.P.); mjsilva@isa.ulisboa.pt (M.J.S.); paula.scotti@iniav.pt (P.S.C.); 3PlantStress & Biodiversity Lab, Centro de Estudos Florestais (CEF), Instituto Superior Agronomia (ISA), Universidade de Lisboa (ULisboa), Quinta do Marquês, Av. República, 2784-505 Lisboa, Portugal; anadr@isa.ulisboa.pt; 4LIBPhys-UNL, Departamento de Física, Faculdade de Ciências e Tecnologia, Universidade NOVA de Lisboa, 2829-516 Caparica, Portugal; mguerra@fct.unl.pt (M.G.); rg.leitao@fct.unl.pt (R.G.L.); 5INIAV—Instituto Nacional de Investigação Agrária e Veterinária, Quinta do Marquês, 2784-505 Oeiras, Portugal; 6ESEAG-COFAC, Avenida do Campo Grande 376, 1749-024 Lisboa, Portugal

**Keywords:** accumulation of nutrients, calcium, photosynthesis, potato biofortification, *Solanum tuberosum* L.

## Abstract

Calcium is essential for plants, yet as its mobility is limited, the understanding of the rate of Ca^2+^ accumulation and deposition in tissues of tubers, as well as the interactions with other critical nutrients prompted this study. To assess the interactions and differential accumulation of micro and macronutrients in the tissues of tubers, *Solanum tuberosum* L. varieties Agria and Rossi were cultivated and, after the beginning of tuberization, four foliar sprayings (at 8–10 day intervals) with CaCl_2_ (3 and 6 kg ha^−1^) or Ca(NO_3_)_2_ (2 and 4 kg ha^−1^) solutions were performed. It was found that both fertilizers increased Ca accumulation in tubers (mostly in the parenchyma tissues located in the center of the equatorial region). The functioning of the photosynthetic apparatus was not affected until the 3rd application but was somewhat affected when approaching the end of the crop cycle (after the 4th application), although the lower dose of CaCl_2_ seemed to improve the photochemical use of energy, particularly when compared with the greater dose of Ca(NO_3_)_2_. Still, none of these impacts modified tuber height and diameter. Following the increased accumulation of Ca, in the tubers of both varieties, the mean contents of P, K, Na, Fe, and Zn revealed different accumulation patterns. Moreover, accumulation of K, Fe, Mn, and Zn prevailed in the epidermis, displaying a contrasting pattern relative to Ca. Therefore, Ca accumulation revealed a heterogeneous trend in the different regions analyzed, and Ca enrichment of tubers altered the accumulation of other nutrients.

## 1. Introduction

Calcium is an essential macronutrient for plant growth and development, providing stability and integrity to the cell wall [1], performing a central role in stress responses [2], and acting as a cofactor of several enzymes involved in the catabolism of ATP and phospholipids [3].

Calcium fluxes from the soil solution, in the form of Ca^2+^, largely occur from the root apex and/or regions of lateral root initiation to the shoot, through the apoplasm (which is relatively non-selective between divalent ions) in regions where Casparian bands are absent, or via the cytoplasm of unsuberized endodermal cells where Casparian bands are present [4,5]. In the apoplasm, cell water space Ca^2+^ binds to negatively charged residues, being taken up by cells down the electrochemical gradient for Ca^2+^, or passing through the water-free space of the cell wall to the xylem [6]. In the xylem, the translocation of Ca^2+^ (or complexed with the organic acid transport pathway) occurs via water mass flow with the transpiration stream [7,8], which determines that Ca transport is inhibited by stomatal closure and that it cannot be readily remobilized in points downstream of the transpirational flow [9]. Indeed, some controversy prevails about Ca redistribution through the phloem, since several authors reported that it is immobile [5,10,11,12], while others pointed out a low kinetics rate [13,14,15,16], namely in potato tubers [17].

To achieve maximum yields, Ca fertilization is required if the soil cannot provide enough nutrients to feed the plant. Nevertheless, Ca fertilization must be adjusted to the final destination because, above the upper threshold, toxic symptoms can develop (namely leaf necrosis, which limits the mobilization of photoassimilates). Besides, unbalanced nutrient interactions that implicate Ca can originate complexes. These complexes can be at the root surface, within the plant organs, and can compete for the sites of uptake, absorption, transport, and function in those tissues (on plant root surfaces or within all plant tissues). In fact, unbalanced nutrients can promote contrasting or similar patterns of interactions with Ca due to the similarities in size, geometry of coordination, and electronic configuration [16,18,19,20,21,22,23,24,25].

As Ca^2+^ is remobilized at a low rate from mature to active growing plant tissues [9], which can ultimately result in local tissue deficits [9], to surpass the imbalance of minerals, foliar spraying can overtake the use of soil as an intermediary for plant nutrition [26]. Moreover, the addition of Ca to the soil can alter tuberization and during this period can also reduce the number of tubers (suppressing tuberization) [27], but the supplemental foliar application of CaCl_2_ and Ca(NO_3_)_2_ at a Ca rate of 0.5–1 kg ha^−1^ (two to four applications, two weeks apart to avoid deficiencies, with the first application occurring at full bloom) is beneficial [28]. Indeed, under this workflow, El-Hadidi et al. (2017) [29] found that, through foliar pulverization, the Ca levels significantly increased in potato leaves and tubers. El-Zohiri and Asfour (2009) [30], through pulverization with Ca(NO_3_)_2_, also found an increased content of Ca in potato tubers, and Seifu and Deneke (2017) [31], spraying with CaCl_2_ and Ca(NO_3_)_2_, obtained a similar trend.

C assimilation is of crucial importance for crop productivity. In fact, photosynthesis is among the first processes affected by environmental constraints, leading to a decline in the growth and productivity of crops, usually to a greater extent when superimposed stresses do occur [32,33,34]. Calcium is implicated in a wide number of physiological processes, including growth and development, as well as tolerance to environmental stresses. Additionally, Ca^2+^ is involved in the regulation of photosynthetic performance, with a wide number of impacts, namely at the levels of stomatal closure, photosystem efficiency, non-photochemical quenching regulation and the xanthophyll cycle [35,36,37]. Fluorescence analysis is a fast, non-destructive, and accurate method to evaluate the functioning of the photosynthetic pathway, including the photochemical use of energy and the photoprotective dissipation processes, and has been extensively used to assess the performance of the photosynthetic apparatus under stressful environmental conditions [34,38,39,40].

Considering previous research [17,41,42], as well as the low kinetic mobility of Ca in the phloem and the limited redistribution in plants, Agria and Rossi varieties were used as a test system to assess the interactions of foliar fertilization with increasing concentrations of CaCl_2_ and Ca(NO_3_)_2_ and the accumulation of mineral elements (Ca, P, K, Na, Fe, Zn, C, H, O, Mn and Cu) in the different tissues of tubers.

## 2. Results

### 2.1. Photosynthetic Apparatus Performance

Within the scope of the functioning of the photosynthetic apparatus, it was found that after the 3rd Ca application, no significant differences were observed in all parameters, regardless of treatment, as compared to the control (Figure 1 and Figure 2). However, after the 4th application, the PSII photochemical efficiency (F_v_/F_m_, F_v_′/F_m_′) was usually reduced only in the Ca(NO_3_)_2_ treatments. At this time, the use of energy through photochemistry (q_L_, Y_(II)_) was not affected as compared with the control. Additionally, the photoprotective dissipation processes (Y_(NPQ)_, q_N_) followed the opposite trend (Figure 1 and Figure 2).

As regards the comparison between the two dates of analysis, a global decline in the use of energy through photochemistry was observed (q_L_, Y_(II)_), likely to be associated with plants approaching the end of their life cycle. However, at this latter developmental stage, some relevant differences were observed regarding the applied Ca products and doses. In fact, the 3 and 6 kg ha^−1^ CaCl_2_ doses promoted the maintenance of F_v_/F_m_ and F_v_′/F_m_′, and somewhat enhanced photosynthetic performance (q_L_, Y_(II)_), denoting as well a marginally lower need of dissipation mechanisms (q_N_, Y_(NPQ)_), always as compared with control plants. In contrast, although q_L_ was not negatively affected by Ca application, and the highest Ca(NO_3_)_2_ treatment (4 kg ha^−1^) reduced the actual PSII photochemical efficiency (F_v_′/F_m_′) and photochemical use of energy (Y_(II)_), the latter, was accompanied by an increase in the photoprotective thermal dissipation of energy (Y_(NPQ)_), although Y_(NO)_ remained largely unchanged regardless of date and applied Ca products/doses, thus reflecting an absence of aggravated non-regulated dissipation processes, that is, showing an absence of photoinhibition.

### 2.2. Mineral Content in Tubers

Compared to the control, in tubers of Agria, the average value of Ca increased significantly, with values ranging between 5–40% (Figure 3). Relative to the control, in tubers of Rossi, Ca content also showed 1.35 and 1.40 fold increases (maximum levels with CaCl_2_—6 kg ha^−1^ and Ca(NO_3_)_2_—4 kg ha^−1^) (Figure 3).

In tubers of Agria and Rossi, the levels of P among the different treatments did not vary significantly, yet values varied between 0.588–1.080% and 0.433–0.820% (with the control mostly showing the lowest values) (Table 1). In the controls of Agria and Rossi, for both Ca foliar fertilizers, the content of K in the tubers showed the lowest values; however, whereas Agria had significantly higher values for all the remaining treatments, Rossi only displayed significantly higher levels through application of CaCl_2_ (Table 1). In the tubers of Agria, Na content revealed the maximum values in the control and minimum values through fertilization with CaCl_2_—6 kg ha^−1^, while increasing concentrations of Ca(NO_3_)_2_ promoted significant and progressive decreases (Table 1). Concerning tubers of Rossi, the amount of Na followed a similar trend after application of increasing concentrations of Ca(NO_3_)_2_, but CaCl_2_ revealed the minimum value through application of CaCl_2_—6 kg ha^−1^ (Table 1). The content of Fe in Agria and Rossi tubers did not vary significantly among treatments (except in CaCl_2_—6 kg ha^−1^), but Zn showed the highest concentration in the Agria control, whereas in the remaining treatments, significant variations were not found (Table 1). Moreover, in tubers of Rossi, the content of Zn did not vary significantly among treatments (Table 1).

### 2.3. Tissue Localization of Mineral Elements in Tubers

Considering that Mg content remained similar among treatments (data not shown), the ratio of Ca/Mg from tubers of Agria and Rossi revealed that Ca prevailed in the center of the equatorial region of the parenchyma tissue (Figure 4).

Although in each treatment significant differences could not be found for C, H and, O contents among tissues from the epidermis to the center of Agria and Rossi tubers (Table 2), in general, the application of CaCl_2_ promoted the lowest values (except in Rossi, treatments of 3 kg ha^−1^ for C and O, and of 6 kg ha^−1^ for C). Additionally, among treatments of both varieties, each region of the tubers displayed similar contents of C, H and O (except in the middle region under treatment with 4 kg ha^−1^ of Ca(NO_3_)_2_).

In each treatment, the control tubers of both varieties revealed the lowest values of K in the epidermis. After Ca spraying with both fertilizers, in each treatment of Agria, a slight increase in K was found (in most cases non-significant) from the epidermis to the center of the tubers. In tubers of Rossi, significant differences were found, but a clear tendency was not observed (Table 2). In the different treatments of both varieties, the lowest content of K occurred in the epidermis (except in the treatment with 4 kg ha^−1^ Ca(NO_3_)_2_); however, for the other regions, a clear trend could not be found (Table 2). Nevertheless, in the center of the tubers, the values of K increased in Agria with CaCl_2_ treatments but decreased in Rossi. Moreover, in the middle and center regions of Rossi, decreases were found with 4 kg ha^−1^ Ca(NO_3_)_2_ treatment, and an increase in the center with 2 kg ha^−1^ of Ca(NO_3_)_2_ treatment (Table 2).

In each treatment of Agria and Rossi, the concentrations of Fe in the tubers remained significantly higher in the epidermis (except for Rossi with 6 kg ha^−1^ CaCl_2_). Among treatments of Agria, the middle region of tubers showed the highest value in the control, whereas the center displayed similar values (Table 2). Concerning the treatments of Rossi, the content of Fe did not vary in the center of tuber, but significantly higher values were found with the 3 kg ha^−1^ CaCl_2_ treatment (Table 2).

In each treatment of Agria, the levels of Mn in the tubers also showed the highest values in the epidermis, but in Rossi a clear trend could not be found (Table 2). Nevertheless, among treatments, a clear trend for Mn accumulation could not be found in the epidermis of Agria, but in Rossi treated with Ca(NO_3_)_2_, an increase was observed (Table 2). In the middle region of tubers from Agria, a decrease in Mn was found with Ca(NO_3_)_2_, but an increase was detected in the center of the parenchyma tissues with CaCl_2_ treatments (Table 2). In the center of the parenchyma tissue of Rossi, relative to the control, Mn content further increased with Ca(NO_3_)_2_ treatments but decreased through the application of CaCl_2_ (Table 2).

In each treatment of Agria, relative to the epidermis, the amount of Zn showed higher values in the center of the tubers (except with the 4 kg ha^−1^ Ca(NO_3_)_2_ and 6 kg ha^−1^ CaCl_2_ treatments), but in Rossi, although in the control a similar tendency occurred, in the remaining treatments, the opposite trend was found (except with 2 kg ha^−1^ Ca(NO_3_)_2_) (Table 2). Among treatments of both varieties, relative to the control, the levels of Zn increased in the epidermis with the Ca(NO_3_)_2_ or CaCl_2_ treatments (Table 2). Among treatments, in the middle region of the tubers from Agria, relative to the control, the content of Zn did not vary significantly with Ca(NO_3_)_2_, but in Rossi, a decrease was found (Table 2). Moreover, in the center of the parenchyma of Agria, a significant increase in Zn was found with CaCl_2_ treatments, but a significant decrease was detected in Rossi treated with either of these fertilizers (Table 2).

In each treatment, the content of Cu did not reveal a clear trend between the epidermis and the center of the tubers (Table 2). Among treatments of Agria, relative to the control, the amount of Cu in the epidermis increased significantly, whereas in the middle region of the tubers, it showed the minimum and higher values with 4 kg ha^−1^ Ca(NO_3_)_2_ and 6 kg ha^−1^ CaCl_2_, respectively (Table 2). In the center of the Agria tubers, minimum values were also found with the 4 kg ha^−1^ Ca(NO_3_)_2_ treatment (Table 2). Among treatments, the Rossi control showed minimum values of Cu in all tuber regions, and among the remaining treatments the Cu content in the center of the parenchyma did not vary significantly (Table 2). Moreover, in the epidermis, treatments with the highest concentrations of Ca(NO_3_)_2_ and the lowest of CaCl_2_ had the highest concentrations of Cu (Table 2). Among all treatments with both fertilizers, the middle region of the parenchyma tubers displayed the highest Cu content with 3 kg ha^−1^ CaCl_2_ (Table 2).

### 2.4. Size and Colorimetric Characteristics

The height and diameter of tubers from both varieties did not vary significantly among treatments (mean values ranged between 9.30–11.80 cm and 4.50–6.24 cm, respectively—Table 3). Regarding the color, parameters L, a* and b* values of tubers of Agria did not vary significantly among treatments. Moreover, in Rossi, parameter a* (red—green transitions) showed significantly higher values through the application of CaCl_2_ relative to the Ca(NO_3_)_2_ treatments and control.

## 3. Discussion

### 3.1. Impact on the Performance of the Photosynthetic Apparatus

Plant species are frequently subjected to an imbalance of nutrients, which adversely affects several metabolic processes, namely those associated with the synthesis of photoassimilates and, therefore, productivity [36,37]. The increase in external Ca concentrations through foliar spraying can modify the photosynthetic performance by reducing stomatal aperture associated with the Ca^2+^-sensing receptor (CAS) protein [43]. This is in line with the significant g_s_ decline previously observed in *S. tuberosum* var. Agria under the same Ca biofortification treatments, although that usually promoted only a (consistent) tendency to reduce the net C assimilation rate after the 3rd application [17]. Calcium is also involved in the cyclic electron flow and non-photochemical quenching [36,37]. In this context, our findings showed (Figure 1 and Figure 2) that after the 3rd application, no impacts were observed regarding the photochemical performance of the photosynthetic machinery or the photoprotective dissipation processes, despite the already mentioned tendency to lower photosynthetic rates [17]. This means that until this stage, the potential functioning of the photosynthetic apparatus was preserved, irrespective of the applied Ca form and doses (Figure 1 and Figure 2), but a marginal impact was observed in C assimilation. The latter was associated with a large decline in g_s_ that restricted the CO_2_ supply (as reflected in the lowered C_i_) to the carboxylation sites [17]. However, after the 4th application of CaCl_2_, the photosynthetic performance was maintained or marginally improved, in contrast with the impact of the Ca(NO_3_)_2_ doses, mainly the highest one. Both Ca(NO_3_)_2_ doses reduced to some extent the PSII photochemical efficiency (F_v_/F_m_, F_v_′/F_m_′), but the highest one additionally reduced the photochemical use of energy (Y_(II)_), with a concomitant increase in the thermal dissipation processes (Y_(NPQ)_). These results suggested a negative impact on the photosynthetic apparatus performance, although without increases in deregulated energy dissipation usually associated with photoinhibition (Y_(NO)_) [40]. Nevertheless, the application of Ca solutions had a neutral effect, considering the absence of significant negative changes in the size of tubers (Table 3) and in the tuber yield of both varieties (data not shown), which indicates that the threshold of toxicity was not reached.

### 3.2. Calcium Accumulation and Interaction with Other Nutrients

After tuberization, Ca accumulation in tubers occurred through foliar spraying with both foliar fertilizers (Figure 3). Agria and Rossi varieties showed, relative to the control, similar upper build-up indexes in the higher treatments (Figure 2). Nonetheless, independently of the accumulation extent of Ca in the potato tubers of both varieties, our data also reinforced that Ca^2+^ mass flow translocated from roots, in the xylem, through the transpiration stream [6,8,10,17,44,45] was accompanied by phloem redistribution of Ca [13,14,15] provided by foliar spraying.

Moreover, the accumulation of Ca determined relevant changes in the accumulation of nutrients and distribution among tissues of tubers. Although in several plant species Ca might stimulate the absorption of P under defined concentration ranges of ions [20,46], at harvest, Ca accumulation promoted by both fertilizers in the tubers of both varieties did not significantly affect the content of P (Table 1). These data indicate the absence of competitive interactions through the development of a linkage bridge between root respiration, ion affinities, and precipitation of calcium phosphates. In fact, during root respiration, hydrogen carbonate ions might keep Ca ions away from the root growing points, whereas uptake and translocation of phosphate from the soil to these growing points secure the uptake and supply of Ca. Thereafter, accumulation of Ca ions at root surfaces may precipitate phosphates and thereby hinder uptake of not only phosphate but also of Ca at harvest [47]. Moreover, Ca accumulation in tubers of both varieties increased K content (although not significantly in Rossi with Ca(NO_3_)_2_), providing evidence of a similar accumulation pattern (Table 1). Thus, our data strongly point out that, as Ca and K have somewhat similar chemical properties (size, charge, geometry of coordination and electronic configuration [48]), competition prevails for the same sites of transport within the tissues of tubers [46]. This trend in potato tubers might link intracellular Ca levels, determining that Ca channels located within the root epidermis and root hair zone can be activated by hyperpolarization of plasma membrane controlling K content through Ca sensing, namely calmodulin, calmodulin-like protein, calcium-dependent protein kinase, and calcineurin B-like protein [25]. In addition, as external Ca^2+^ alters the selectivity of nonselective cation channels, favoring the uptake of K^+^, which is one of the main competitors of Na^+^ entrance into the roots [49], our data suggested the development of a contrasting interaction for Na^+^ accumulation in both varieties (Table 1). Indeed, although salinity reduces Ca^2+^ uptake and translocation, the increasing supply of Ca to both potato varieties ameliorated the deleterious effects of Na^+^ [50,51], favoring plant growth [52,53]. Micronutrients such as Fe and Zn play very important roles in the physiological processes of plant species [54,55,56]. Nevertheless, although in some plant species a contrasting interaction prevails between Ca or K and Fe accumulations [24,57,58], it was interesting to notice that the increasing contents of Ca and K in the potato tubers did not determine significant changes in Fe in all treatments of each variety (except with 6 kg ha^−1^ Ca(NO_3_)_2_) (Table 1). Thus, according to our data, potato tubers pointed to the absence of relations of a competitive nature between Ca and Fe, eventually driven by isolated or combined interactions with these macronutrients. Additionally, Zn content also did not vary significantly through the application of both fertilizers in Rossi tubers (Table 1); however, as previously found in soybean and wheat [24], relative to the control, in Agria tubers, the significantly lower levels of Zn mediated by both fertilizers suggested the potential occurrence of a contrasting pattern of interaction with Ca (Table 1). However, following a multi-level nutritional interaction approach, it is known that Zn deficiency induces the expression of several P assimilation-related genes [59], while P deficiency activates the expression of the genes involved in Zn and Fe homeostasis [60,61]. Thus, in Ca-treated tubers of both varieties P, Fe, and Zn in general did not vary significantly; it can be further assumed that the mechanisms involved in coordinating these interactions of nutrients and nutrient-stress signaling were not activated by Ca.

### 3.3. Accumulation in Tuber Tissue

The higher accumulation of Ca in the center of tubers of both varieties was related to the low translocation rate of Ca to the tubers after the application of both fertilizers (Figure 3). Indeed, a high translocation rate of Ca^2+^ (or complexed with the organic acid transport pathway) was driven in the xylem by water mass flow via the transpiration stream [7,8], to the areal part of the plants. Moreover, once uploaded from the xylem, and in particular to points downstream of transpirational flow, Ca transport cannot be readily remobilized [9] to the tubers and diffused to the peripheral regions.

The differential accumulation of Ca in tubers of both varieties, through foliar pulverization with CaCl_2_ and Ca(NO)_3_)_2_, did not determine significant variations in C, H, and O among tissues (Table 2). Moreover, in terms of the similar heights and diameters of tubers (Table 3), our data pointed to the absence of inhibitory interactions between Ca and these chemical elements, despite the impact observed in some photosynthesis-related parameters after the 4th Ca(NO_3_)_2_ treatment (Figure 1 and Figure 2). Indeed, C, H, and O are building blocks for proteins, lipids, starch, and other carbohydrates, lignins, and celluloses, mediated by the synthesis and mobilization of photoassimilates, determining the rate of plant growth.

The lower values of K accumulation in the epidermis of tubers (Table 3) suggest a predominant participation of this nutrient in the inner parts of the tubers, where it determines the metabolic accumulation of phytonutrients or bioactive compounds [62,63]. Indeed, unlike other nutrients, K is not incorporated into structures of organic compounds, remaining in an ionic form (K^+^), in a solution of cells, and has many functions in plant nutrition and growth that influence both yield and quality. These include the facilitation of cell division and growth through participation in the mobilization of starches and proteins between plant parts.

The contrasting pattern between Ca and Fe accumulation among tissues of tubers in all treatments (epidermis and the central region of the tubers, respectively—Table 2; Figure 4) followed a pattern previously found in other plant species [64] for both Fe uptake and transport between plant organs. Indeed, Fe uptake prevails in the root tips and is metabolically controlled, its translocation being largely mediated by Ca citrate chelates [4,64,65].

Uptake and translocation of Mn, like Ca, is also metabolically controlled [65,66,67,68], yet the differential concentrations in the tissues of tubers from both varieties (i.e., higher levels of Mn and Ca in the epidermis and the parenchyma tissues in the center, respectively—Table 2; Figure 4) also suggest a contrasting pattern of accumulation between these chemical elements. Indeed, it was interesting to notice a similar pattern, at organ levels, among these nutrients in other plant species [69,70]. Moreover, the absence of a clear trend of Mn accumulation within each treatment additionally suggests the parallel occurrence of a passive transport within tissues. Indeed, generally, Mn is known to be rapidly transported within plant tissues without being bound to insoluble organic ligands [65].

Root uptake of Zn is mostly metabolically controlled (although a non-metabolic process might also occur), having high translocation rates in the form of both hydrated Zn and Zn^2+^, eventually also bound to light organic compounds [65,71]. In this context, the increasing accumulation of Zn in the epidermis of Ca-treated tubers in most treatments (4 kg ha^−1^ Ca(NO_3_)_2_ and 6 kg ha^−1^ CaCl_2_ for both varieties and in the 3 kg ha^−1^ CaCl_2_ treatment for Rossi), in parallel with the lowest levels of Ca (Table 1; Figure 3), suggests an increasing rate of passive Zn absorption. Moreover, the decreasing accumulation of Zn from the epidermis to the center of tubers in each Ca treatment (except for the control and the 2 kg ha^−1^ Ca(NO_3_)_2_ treatment in both varieties and the 3 kg ha^−1^ CaCl_2_ treatment in Agria), seems to indicate a contrasting pattern of accumulation, prevailing in the center of tubers (Figure 3). In fact, this interaction was previously found in the organs of several plant species [46].

Depending on plant species, active or passive Cu uptake mechanisms can prevail, conditioning the metabolic pathways [65,72,73]. In this context, the interactions between Cu and Ca are highly complex and apparently are cross-linked with the accumulation kinetics of other nutrients in the different regions of the tubers. Nevertheless, the well-known affinity of carbonates to precipitate Cu and its relatively low mobility within root cells [65,73,74] seem to determine a higher accumulation of Cu in the epidermis, whereas Ca prevails in the center of tubers among treatments and in both varieties (Table 1; Figure 3).

It has long been known that differential mineral concentrations might be causing variations in color intensity or different coloring in potato tubers, namely black by ferrous compounds, discoloration by K or Mg deficiency, and reddish-brown color mediated by Cu treatments [75,76]. Nevertheless, it was interesting to notice that Ca, directly or through interactions with other nutrient accumulations, or via differential deposition in the tissues of potato tubers, did not promote color changes in either variety treated with either fertilizer (except in the a* parameter for the Rossi variety).

## 4. Materials and Methods

### 4.1. Experimental Fields

*Solanum tuberosum* L. varieties Agria and Rossi were cultivated in a potato production region of the west of Portugal (GPS coordinates 39°16′38.77″ N; 9°15′8.294″ W for Agria variety; 38°16′31.76″ N; 9°13′46.77″ W for Rossi variety). Both fields were situated 133 m above sea level, having soils with pH of 7.4, a clay loam texture and an electrical conductivity of 205 and 349 µS cm^−1^ and 0% and 4% of carbonates (for Agria and Rossi, respectively). The contents of nutrients in the soils for Agria and Rossi cultivation were: 0.39% and 0.71% of Ca, 2.20% and 2.64% of K, 0.15% and 0.24% of Mg, 0.23% and 0.19% of P, 1.19% and 0.50% of Fe, 55.9 and 66.6 ppm of S, 19.6 and 41.7 ppm of Zn, and 318 and 270 ppm of Mn, respectively. Additionally, organic matter content was 1.18% and 4.13% for the Agria and Rossi fields, respectively. Under adequate irrigation conditions (provided by sprinkler systems according to weather conditions), the production of tubers took place from 4 May to 24 September 2018. Maximum and minimum average air temperatures were 23 °C and 15 °C (with maximum and minimum values of 41 °C and 6 °C, respectively), respectively. The average rainfall was 0.41 mm, with a daily maximum of 18.03 mm and an accumulation of 60.4 mm. There were no periods of rain after any of the foliar applications in both fields.

After the beginning of tuberization (in the beginning of July for Agria and end of July for Rossi), four foliar sprayings with CaCl_2_ (3 and 6 kg ha^−1^) or Ca(NO_3_)_2_ (2 and 4 kg ha^−1^) solutions were performed (Table 4).

Control plants were sprayed with water. The experimental plots for Agria and Rossi (Figure 5) contained all treatments (control included), having been carried out in quadruplicate (compass, 60–80 cm).

### 4.2. Chlorophyll a Fluorescence Parameters

Agria plants were used as a test system to monitor leaf chlorophyll (Chl) a fluorescence parameters evaluated after the 3rd (7 days after the foliar application) and 4th (11 days after the foliar application) leaf applications. The analysis was carried out using a PAM-2000 system (H. Walz, Germany) as previously described [40], following the formulae discussed elsewhere for calculations [38,77,78]. Measurements were carried out in 5 independent leaves of 5 different plants per treatment. Measurements of minimal fluorescence (F_0_), maximal fluorescence (F_m_), and maximal photochemical efficiency of photosystem (PS) PSII (F_v_/F_m_), were performed on overnight dark-adapted leaves. F_0_ was assessed using a weak light (<0.5 μmol m^−2^ s^−1^) beam, while F_m_ was obtained using a saturation flash of ca. 7500 μmol m^−2^ s^−1^ of actinic light for 0.8 s.

Another group of parameters were determined under photosynthetic steady-state conditions (with at least 3 to 4 h of light exposure), under natural irradiance (ca. 1200–1400 μmol m^−2^ s^−1^) and superimposed saturating flashes: F_v_′/F_m_′, q_L_, q_N_, Y_(II)_, Y_(NPQ)_, Y_(NO)_ [38,66,67]. F_0_′, which was required for the quenching calculations, was measured in the dark, immediately after the actinic light was switched off and before the first fast phase of the fluorescence relaxation kinetics. F_v_′/F_m_′ expresses the PSII photochemical efficiency of energy conversion under light exposure. q_L_ is the photochemical quenching based on the concept of interconnected PSII antennae, and represents the proportion of energy captured by open PSII centers and driven to photochemical events. Estimates of photosynthetic quantum yields of non-cyclic electron transfer (Y_(II)_), photoprotective regulated energy dissipation of PSII (Y_(NPQ)_), and non-regulated energy dissipation (heat and fluorescence) of PSII (Y_(NO)_) were also calculated, where Y_(II)_ + Y_(NPQ)_ + Y_(NO)_ = 1.

### 4.3. Analysis of Total Nutrients by Atomic Absorption Spectrophotometry

After harvest in the experimental potato-producing fields, eight randomized samples of tubers (of similar size) from each treatment were washed, dried at 60 °C until constant weight, and ground in an agate mortar. The homogenates were further divided into four samples (*n* = 4), and an acid digestion procedure was performed with a mixture of HNO_3_-HClO_4_ (4:1) according to [79,80]. After filtration, total Ca, P, K, Na, Fe and Zn contents were measured in triplicate by atomic absorption spectrophotometry, using a model Perkin Elmer AAnalyst 200 (Waltham, MA, USA), and the absorbance in mg/L was determined with coupled AA WinLab software (version 32) program.

### 4.4. Analysis of Ca/Mg by Scanning Electron Microscope Coupled to X-ray Dispersive Energy Spectroscopy

At harvest, the tubers from all treatments, after washing with deionized water, were cut transversely at the equatorial region, and the slices dehydrated and dried in CO_2_, using a Balzers Union CPD 020 system. Samples thereafter were adhered to 13 mm aluminum stubs with conductive carbon adhesive pads and sputter-coated with gold to an approximate thickness of 10–15 nm, using a Polaron equipment coating unit. The Ca/Mg ratio was then determined using a scanning electron microscope (SEM) JEOL JSM-T330A model, coupled to an X-ray dispersive energy spectroscopy (EDS) device (acceleration voltage of 25 kV, beam current of 4–6 mÅ, 200 s pre-set time for spectrum acquisition, 2048 channels, 10 eV/channel, 20 keV width, Si crystal detector, protective window with Al coating) and a Tracor Northern Series II microanalyzer (Series TN5502N EDS System). To capture images, the AnalySIS 3.0 version software from Soft Imaging System GmbH (Munster, Germany) was used. The semi-quantitative chemical analyses were performed using the Quest SpectraPlus software from Thermo Noran (Karlsruhe, German). Each measurement was carried out in quadruplicate on four different potatoes cut transversely.

### 4.5. Analysis of Nutrients in Tissues through Fluorescence Detection

Slices with a width of 4 mm (three replicates from three independent series) from harvested tubers from all treatments were washed with deionized water, cut transversely at the equatorial region, and dried at 60 °C. Quantification of C, H, O, K, P, Fe, Mn, Zn and Cu was further carried out (four measurements per treatment in three regions of the tubers, between the peel and the core) following [81]. A µ-EDXRF system (M4 Tornado™, Bruker, Germany) was used. The X-ray generator was operated at 50 kV and 100 µA without the use of filters, to enhance the ionization of low-Z elements. All of the measurements with filters were performed with 600 µA voltage. Detection of fluorescence radiation was performed by an energy-dispersive silicon drift detector, XFlash™, with 30 mm^2^ sensitive area and energy resolution of 142 eV for Mn Kα. Measurements were carried out under 20 mbar vacuum conditions. These point spectra were acquired during 200 s. The measurements were performed in three replicates of three independent series.

### 4.6. Height, Diameter, and Colorimetric Parameters

Height and diameter were measured considering 10 randomized tubers per treatment (10 samples from each treatment) and from three independent plant series (10 × 10 × 3 measurements). The measurement of colorimetric parameters, using fixed wavelength, was carried out according to [82]. Brightness (L) and chromaticity parameters (a* and b* coordinates) were obtained with a Minolta CR 300 colorimeter (Minolta Corp., Ramsey, NJ, USA) coupled to a sample vessel (CR-A504). The system of the Commission Internationale de I’Éclaire (CIE) was applied using the illuminant D65. Parameter L represents the brightness of the sample, indicating the variation in the tonality between dark and light (range between 0—black and 100—white). Parameters a* and b* indicate color variations between red (+60) and green (−60), and between yellow (+60) and blue (−60), respectively. The null value approximation of these coordinates indicates neutral colors such as white, gray, and black. Measurements were carried out at harvest, in triplicates from three independent series.

### 4.7. Statistical Analysis

Data were statistically analyzed using one-way or two-way ANOVA (*p* ≤ 0.05) through IBM SPSS software, to assess differences between treatments and experimental periods and, based on the results, a Tukey’s test for mean comparison was performed, considering a 95% confidence level.

## 5. Conclusions

During tuberization, foliar spraying with CaCl_2_ or Ca(NO_3_)_2_ at maximum concentrations of 6 kg ha^−1^ and 4 kg ha^−1^, respectively, increased Ca accumulation in tubers (mostly in the parenchyma tissues located in the center of the equatorial region) of Agria and Rossi varieties. The Ca applications promoted minor impacts in the C assimilation pathway until the 3rd application, likely associated with a CO_2_ restriction to the carboxylation sites due to a clear g_s_ decline (in all Ca treatments). In tubers of both varieties, Ca accumulation promoted a similar pattern of accumulation with K, as well as the absence of competitive interactions with P and Fe and a contrasting pattern relative to Na. Moreover, the interaction between Ca and Zn seemed to depend on the potato variety, since through application of both fertilizers, Zn content did not vary significantly in Agria, but a potential contrasting pattern occurred in Rossi. Regarding the tissues of tubers, the accumulation of K, Fe, Mn, and Zn prevailed in the epidermis, displaying a contrasting pattern relative to Ca. Moreover, the interaction between Ca and Cu accumulation was apparently cross-linked with the accumulation kinetics of other nutrients in the different regions of the tubers. Additionally, CaCl_2_ or Ca(NO_3_)_2_ applications did not alter colorimetric parameters in either variety.

## Figures and Tables

**Figure 1 plants-11-01725-f001:**
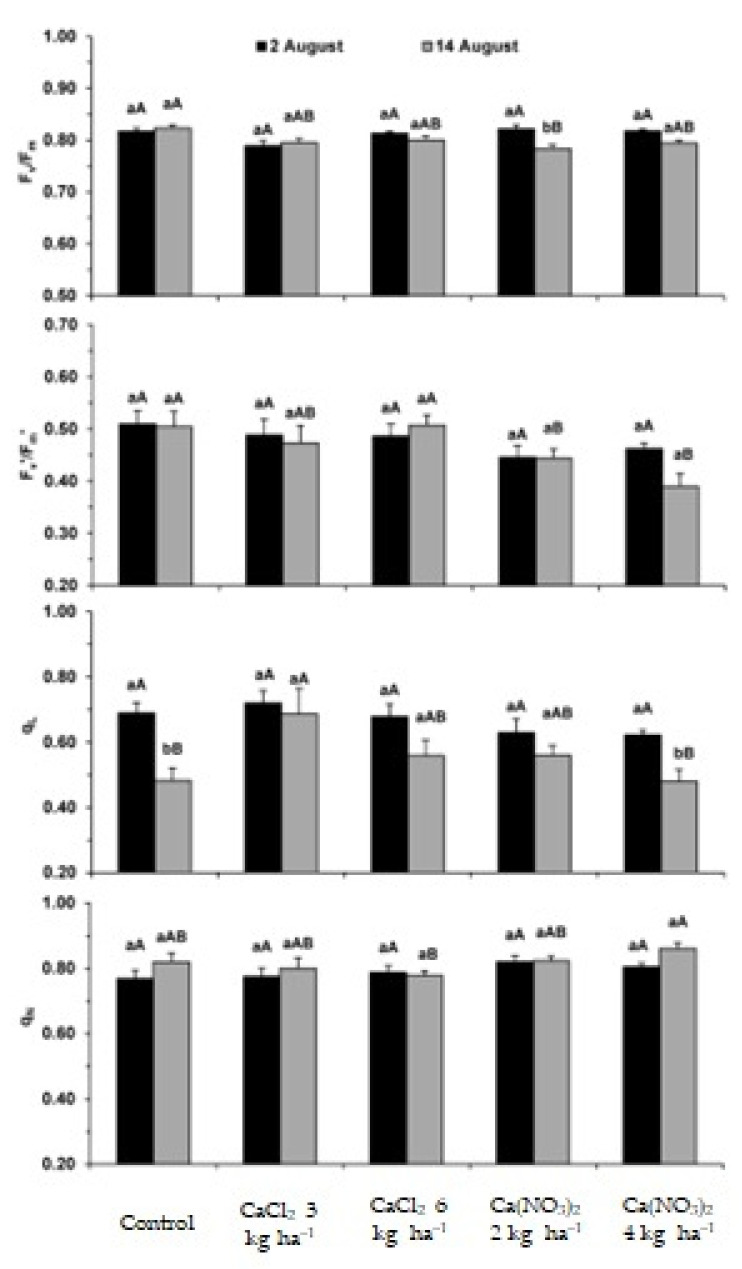
Leaf chlorophyll a fluorescence parameters in *S. tuberosum* L. cv. Agria after the 3rd (2 August) and 4th (14 August) calcium applications. Parameters include the maximum (F_v_/F_m_) and actual (F_v_′/F_m_′) PSII photochemical efficiency, photochemical quenching coefficient (q_L_), and non-photochemical quenching (q_N_). For each parameter, the different letters after the mean values ± S.E. (*n* = 5) express significant differences between Ca treatments within each date (A, B), or between dates for each Ca treatment (a, b).

**Figure 2 plants-11-01725-f002:**
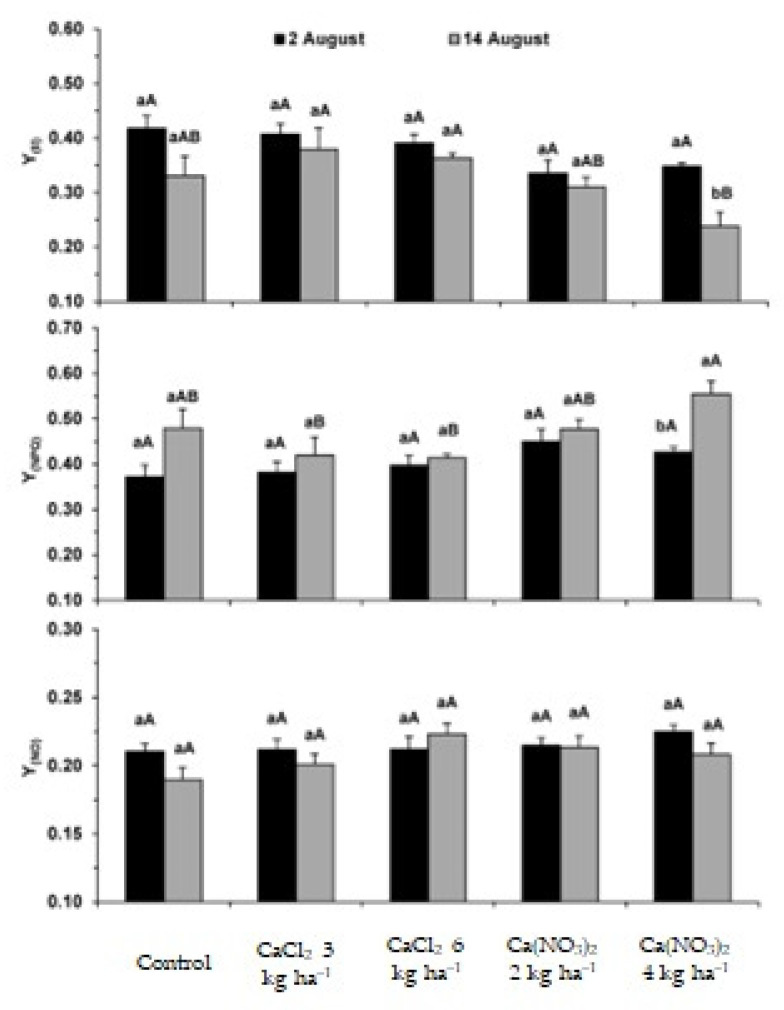
Leaf chlorophyll a fluorescence parameters in *S. tuberosum* L. cv. Agria after the 3rd (2 August) and 4th (14 August) calcium applications. Parameters include: the estimate of quantum yields of non-cyclic electron transport (Y_(II)_) of regulated energy dissipation in PSII (Y_(NPQ)_) and of non-regulated energy dissipation in PSII (Y_(NO)_). For each parameter, the different letters after the mean values ± S.E. (*n* = 5) express significant differences between Ca treatments within each date (A, B), or between dates for each Ca treatment (a, b).

**Figure 3 plants-11-01725-f003:**
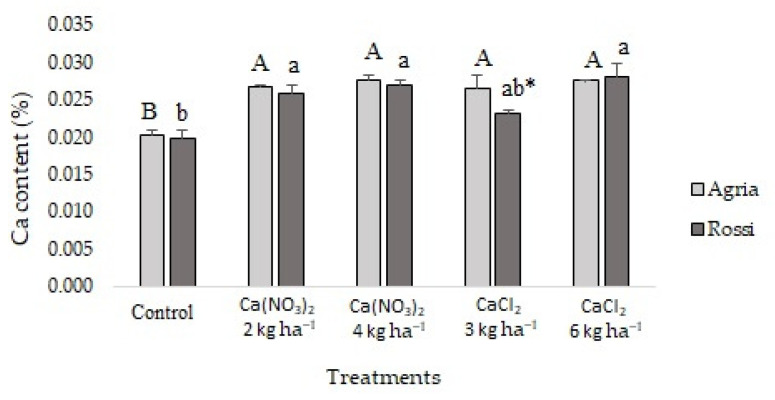
Calcium concentration in *S. tuberosum* L. cv. Agria and Rossi at harvest, submitted to four foliar spraying applications (at 8–10 day intervals) of CaCl_2_ (3 and 6 kg ha^−1^) or Ca(NO_3_)_2_ (2 and 4 kg ha^−1^). Different letters after the mean values ± S.E. (three replicates of three independent series) express significant differences between Ca treatments in Agria variety (A, B), or between Ca treatments in Rossi variety (a, b). There was only a significant difference in Ca content between Agria and Rossi varieties in CaCl_2_ (3 kg ha^−1^), with Rossi content being significantly lower (*).

**Figure 4 plants-11-01725-f004:**
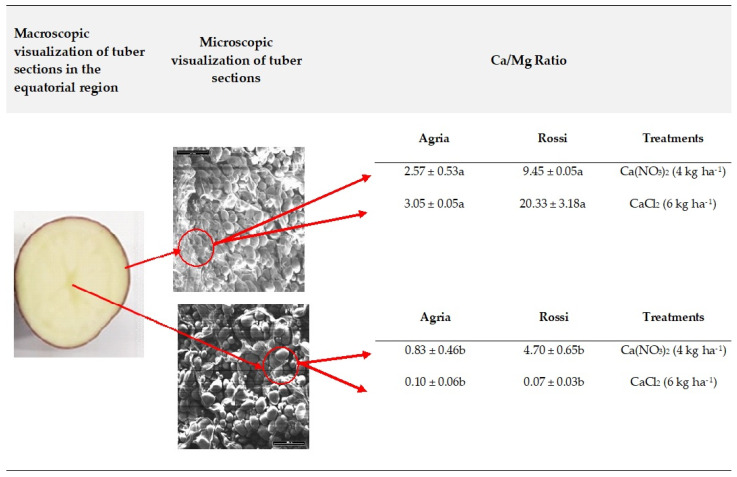
Ca/Mg ratio in the epidermis and center, in the equatorial region of tubers of *S*. *tuberosum* L. cv. Agria and Rossi at harvest, after four foliar sprayings with Ca(NO_3_)_2_ (4 kg ha^−1^) and CaCl_2_ (6 kg ha^−1^). The mean values (±S.E.) of Mg in Agria after spraying with Ca(NO_3_)_2_ (4 kg ha^−1^) and CaCl_2_ (6 kg ha^−1^) were 192 ± 8 and 192 ± 2 ppm, respectively, and in Rossi were 194 ± 4 and 194 ± 6 ppm, respectively. For each variety and treatment, corresponds to the mean of 3 replicates from 3 independent series ± S.E. Letters a, b represent significant differences between the epidermis and center of tubers of each variety and treatment (statistical analysis using the single factor ANOVA test, *p* ≤ 0.05).

**Figure 5 plants-11-01725-f005:**
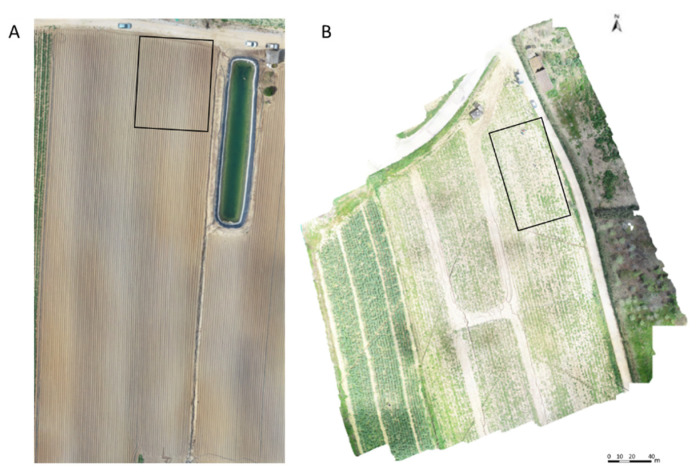
Orthophoto maps of the fields in which the varieties were implemented and the biofortification treatments were carried out (Agria—(**A**); Rossi—(**B**)). The images were obtained by UAV on 18 May 2018 and 12 May 2018, respectively, for fields A and B.

**Table 1 plants-11-01725-t001:** Mineral element concentrations in *S. tuberosum* L. cv. Agria and Rossi at harvest, submitted to four foliar sprayings (at 8–10 day intervals) with CaCl_2_ (3 and 6 kg ha^−1^) or Ca(NO_3_)_2_ (2 and 4 kg ha^−1^). Mean values ± S.E. (three replicates of three independent series).

Treatments		P	K	Na	Fe	Zn
		%	%	ppm	ppm	ppm
		Agria				
Control		0.588 ± 0.209a	0.272 ± 0.006c	87.5 ± 1.81a	114 ± 2.97a	53.6 ± 9.29a
Ca(NO_3_)_2_	2 kg ha^−1^	0.845 ± 0.053a	0.312 ± 0.003b	76.6 ± 0.81b	135 ± 3.22a	31.8 ± 1.55b
	4 kg ha^−1^	0.448 ± 0.196a	0.337 ± 0.010ab	69.1 ± 2.79b	134 ± 6.32a	29.9 ± 1.83b
CaCl_2_	3 kg ha^−1^	0.903 ± 0.114a	0.370 ± 0.009a	58.2 ± 1.38c	123 ± 6.52a	33.9 ± 0.81ab
	6 kg ha^−1^	1.08 ± 0.26a	0.368 ± 0.004a	63.9 ± 0.84b	131 ± 5.35a	31.4 ± 3.51b
		**Rossi**				
Control		0.433 ± 0.096a	0.270 ± 0.006b	86.0 ± 2.47a	71 ± 15.8b	26.5 ± 0.39a
Ca(NO_3_)_2_	2 kg ha^−1^	0.820 ± 0.052a	0.279 ± 0.006b	80.9 ± 1.76ab	106 ± 6.13b	27.2 ± 1.39a
	4 kg ha^−1^	0.451 ± 0.064a	0.286 ± 0.007b	71.5 ± 1.57b	107 ± 4.48b	27.6 ± 0.43a
CaCl_2_	3 kg ha^−1^	0.811 ± 0.140a	0.340 ± 0.009a	72.6 ± 1.58b	108 ± 3.31b	26.8 ± 0.72a
	6 kg ha^−1^	0.705 ± 0.284a	0.362 ± 0.016a	39.5 ± 3.25c	183 ± 27.7a	31.0 ± 1.59a

Letters a–c represent significant differences between treatments for each variety (statistical analysis using the single factor ANOVA test, *p* ≤ 0.05).

**Table 2 plants-11-01725-t002:** Chemical element content in different parts of the equatorial region, ranging from the epidermis (1) to the center (3) in tubers of *S*. *tuberosum* L. cv. Agria and Rossi at harvest, submitted to four foliar sprayings (at 8–10 day intervals) with CaCl_2_ (3 and 6 kg ha^−1^) or Ca(NO_3_)_2_ (2 and 4 kg ha^−1^). Mean values ± S.E. (three replicates of three independent series).

Treatments		Regions	C	H	O	K	Fe	Mn	Zn	Cu
			g kg^−1^					mg kg^−1^		
			Agria							
Control		1	419 ± 21Aa	58.6 ± 2.9Aa	465 ± 23Aa	40.1 ± 2.0Bb	1.12 ± 0.10Aa	35.7 ± 1.8Ab	17.5 ± 0.9Bc	5.74 ± 0.29Cc
		2	392 ± 20Aa	54.8 ± 2.7Aa	435 ± 22Aa	74.2 ± 3.7Ab	0.15 ± 0.00Ba	6.10 ± 0.31Cb	22.6 ± 1.1Bb	7.03 ± 0.35Bbc
		3	386 ± 19Aa	54.0 ± 2.7 Aa	429 ± 21Aa	72.4 ± 3.6Ab	0.10 ± 0.00Ca	8.43 ± 0.42Bc	27.9 ± 1.4Ab	9.90 ± 0.49Aab
Ca(NO_3_)_2_	2 kg ha^−1^	1	428 ± 21Aa	59.9 ± 3.0Aa	475 ± 24Aa	27.9 ± 1.4Bc	0.40 ± 0.00Ac	37.23 ± 0.36Ab	12.9 ± 0.7Bc	4.17 ± 0.21Bc
		2	418 ± 21Aa	58.4 ± 2.9Aa	464 ± 23Aa	50.2 ± 2.5Ac	0.10 ± 0.00Bb	3.27 ± 0.16Cc	23.4 ± 1.2Ab	8.01 ± 0.40Ab
		3	404 ± 20Aa	56.4 ± 2.8 Aa	448 ± 22Aa	56.9 ± 2.9Ac	0.10 ± 0.00Ba	10.1 ± 0.5Bbc	30.9 ± 1.5Ab	10.1 ± 0.50Aab
	4 kg ha^−1^	1	400 ± 20Aa	56.0 ± 2.8Aa	445 ± 22Aa	76.2 ± 3.8Aa	1.42 ± 0.10Aa	52.1 ± 2.6Aa	37.4 ± 1.9Aa	9.67 ± 0.48Ab
		2	328 ± 16Bb	45.9 ± 2.3Aa	365 ± 18Aa	85.5 ± 4.3Ab	0.10 ± 0.00Bb	4.41 ± 0.22Cc	25.1 ± 1.3Bb	5.33 ± 0.27Bc
		3	369 ± 19Bab	51.6 ± 2.6 Aa	410 ± 21Aa	72.1 ± 3.6Ab	0.10 ± 0.00Ba	6.71 ± 0.34Bc	18.2 ± 0.9Cc	7.83 ± 0.39Bc
CaCl_2_	3 kg ha^−1^	1	375 ± 19Aa	52.5 ± 2.6Aa	416 ± 21ABa	50.0 ± 2.5Bb	0.50 ± 0.00Ab	22.1 ± 1.1Ac	25.3 ± 1.3ABb	7.45 ± 0.37Bb
		2	416 ± 21Aa	58.3 ± 2.9Aa	462 ± 23Aa	55.1 ± 2.8Bc	0.10 ± 0.00Bb	10.7 ± 0.5Ba	19.7 ± 1.0Bb	8.67 ± 0.43Bb
		3	363 ± 18Aa	50.7 ± 2.5 Aa	402 ± 20Ba	76.0 ± 3.8Ab	0.10 ± 0.00Ba	12.1 ± 0.6Bb	29.8 ± 1.5Ab	13.1 ± 0.70Aa
	6 kg ha^−1^	1	383 ± 19Aa	53.6 ± 2.7Aa	426 ± 21Aa	78.2 ± 3.9Ba	0.80 ± 0.00Ab	38.0 ± 1.9Ab	40.1 ± 2.0Aa	25.0 ± 1.30Aa
		2	363 ± 18Aab	50.8 ± 2.5Aa	403 ± 20Aa	99.2 ± 5.0AABa	0.12 ± 0.01Bb	9.37 ± 0.5Ca	37.6 ± 1.9Aa	26.7 ± 1.30Aa
		3	344 ± 17Aa	49.1 ± 2.4 Aa	382 ± 19Aa	110 ± 50BAa	0.12 ± 0.01Ba	18.3 ± 0.9Ba	37.9 ± 1.9Aa	10.7 ± 0.50Bab
			**Rossi**							
Control		1	425 ± 21Aa	59.5 ± 3.0Aa	472 ± 24Aa	30.4 ± 1.5Cc	0.70 ± 0.03Ab	13.8 ± 0.7Ad	8.77 ± 0.4Cd	n.d.
		2	394 ± 20Aa	55.1 ± 2.8Aa	438 ± 22Aa	65.4 ± 3.3Bb	0.07 ± 0.01Bb	11.5 ± 0.6Ac	23.7 ± 1.2Bb	0.14 ± 0.01c
		3	379 ± 19Aa	53.1 ± 2.7 Aa	421 ± 21Aa	103 ± 5.0Aa	0.06 ± 0.01 Ba	13.2 ± 0.7Ab	35.5 ± 1.8Aa	n.d.
Ca(NO_3_)_2_	2 kg ha^−1^	1	426 ± 21Aa	59.5 ± 3.0Aa	472 ± 24Aa	36.5 ± 1.8Bc	0.63 ± 0.03Ab	28.9 ± 1.4Ac	18.3 ± 0.9ABc	6.42 ± 0.32Bc
		2	420 ± 21Aa	58.8 ± 3.0Aa	467 ± 23Aa	48.5 ± 2.4Bc	0.04 ± 0.01Bb	5.7 ± 0.28Cd	15.6 ± 0.8Bc	6.20 ± 0.31Bb
		3	371 ± 19a	51.8 ± 2.6Aa	411 ± 21Aa	60.0 ± 3.0Ab	0.08 ± 0.01Ba	21.8 ± 1.1Ba	25.6 ± 1.3Aa	9.25 ± 0.46Aa
	4 kg ha^−1^	1	350 ± 18Aa	49.0 ± 2.5Aa	389 ± 19Aa	88.0 ± 4.4Aa	1.34 ± 0.07Ab	71.1 ± 3.6Aa	68.7 ± 3.4Aa	23.00 ± 1.2Ab
		2	370 ± 19Aa	51.7 ± 2.6Aa	410 ± 21Aa	29.4 ± 1.5Bd	0.08 ± 0.01Bb	19.7 ± 1.0Bb	16.3 ± 0.8Bc	5.83 ± 0.29Bb
		3	371 ± 19Aa	51.9 ± 2.6 Aa	412 ± 21Aa	28.1 ± 1.4Bd	0.09 ± 0.01Ba	25.9 ± 1.3Ba	14.3 ± 0.7Bb	7.67 ± 0.38Ba
CaCl_2_	3 kg ha^−1^	1	362 ± 18Aa	50.6 ± 2.5Aa	401 ± 20Aa	80.1 ± 4.0Ba	2.26 ± 0.11Aa	53.2 ± 2.7Ab	61.1 ± 3.1Aa	32.60 ± 1.6Aa
		2	363 ± 18Aa	50.7 ± 2.5Aa	403 ± 20Aa	117.0 ± 6.0Aa	0.15 ± 0.01Ba	57.0 ± 2.9Aa	53.1 ± 2.7Aa	21.10 ± 1.1Ba
		3	390 ± 20Aa	54.6 ± 2.7 Aa	433 ± 22Aa	38.5 ± 1.9Ca	0.06 ± 0.01Ca	8.98 ± 0.45Bc	18.5 ± 0.9Bb	9.07 ± 0.50Ca
	6 kg ha^−1^	1	413 ± 21Aa	57.8 ± 2.9Aa	458 ± 23Aa	51.3 ± 2.6Ab	0.11 ± 0.01Ac	13.0 ± 0.7Ad	25.3 ± 0.0Ab	9.83 ± 0.46Ac
		2	363 ± 18Aa	50.8 ± 2.5Aa	403 ± 20Aa	30.6 ± 1.5Bd	0.09 ± 0.01Aab	4.9 ± 0.25Cd	13.3 ± 4.4Bc	8.61 ± 0.43Ab
		3	382 ± 19Aa	53.4 ± 2.7 Aa	423 ± 21Aa	36.1 ± 1.8Bc	0.10 ± 0.01Aa	8.9 ± 0.44Bc	14.2 ± 0.7Bb	9.22 ± 0.46Aa

Different letters express significant differences (*p* < 0.05) among different tissues of each treatment (A, B, C), or within each tissue from different treatments of the same genotype (a–d). “n.d.” means not detected.

**Table 3 plants-11-01725-t003:** Height, diameter, and colorimeter parameters of the fresh pulp from tubers of *S*. *tuberosum* L. cv. Agria and Rossi at harvest, submitted to four foliar sprayings (at 8–10 day intervals) with CaCl_2_ (3 and 6 kg ha^−1^) or Ca(NO_3_)_2_ (2 and 4 kg ha^−1^). Mean values ± S.E. (10 randomized tubers per treatment (10 samples of each treatment) and from three independent plant series).

Treatments		Height (cm)	Diameter (cm)	Color Parameters		
				L	a*	b*
**Agria**						
Control		9.30 ± 0.19a	5.60 ± 0.09a	54.4 ± 1.31a	−2.89 ± 0.17a	20.5 ± 0.22a
Ca(NO_3_)_2_	2 kg ha^−1^	11.80 ± 0.31a	5.47 ± 0.13a	56.8 ± 0.86a	−2.90 ± 0.13a	21.3 ± 0.49a
	4 kg ha^−1^	11.40 ± 0.91a	4.77 ± 0.15a	59.5 ± 0.36a	−3.31 ± 0.07a	22.3 ± 0.33a
CaCl_2_	3 kg ha^−1^	9.50 ± 0.19a	5.70 ± 0.12a	58.8 ± 0.50a	−2.95 ± 0.21a	22.0 ± 0.29a
	6 kg ha^−1^	9.90 ± 0.68a	4.73 ± 0.28a	55.8 ± 1.29a	−2.85 ± 0.08a	21.7 ± 1.39a
**Rossi**						
Control		9.60 ± 0.32a	6.24 ± 0.15a	65.9 ± 1.19a	−2.52 ± 0.14c	16.3 ± 0.30a
Ca(NO_3_)_2_	2 kg ha^−1^	7.03 ± 0.53a	4.50 ± 0.22a	63.3 ± 0.73a	−2.64 ± 0.12c	14.2 ± 0.27a
	4 kg ha^−1^	9.32 ± 1.58a	5.50 ± 0.26a	63.3 ± 1.12a	−2.75 ± 0.07c	15.7 ± 0.24a
CaCl_2_	3 kg ha^−1^	9.30 ± 0.32a	5.65 ± 0.21a	59.9 ± 2.55a	−2.15 ± 0.09a	15.1 ± 0.77a
	6 kg ha^−1^	8.91 ± 1.17a	5.61 ± 0.49a	61.4 ± 1.88a	−2.34 ± 0.05b	15.1 ± 0.65a

Letters a–c represent significant differences between treatments for each variety (statistical analysis using the single-factor ANOVA test, *p* ≤ 0.05).

**Table 4 plants-11-01725-t004:** Planting, foliar application, and harvest dates in Agria and Rossi varieties.

Varieties	Planting	Foliar Applications				Harvest	Treatments
		1°	2°	3°	4°		
Agria	4 May 2018	6 July 2018	16 July 2018	26 July 2018	3 August 2018	4 September 2018	CaCl_2_ (3 and 6 kg ha^−1^) or, Ca(NO_3_)_2_ (2 and 4 kg ha^−1^)
Rossi	11 May 2018	25 July 2018	3 August 2018	14 August 2018	24 September 2018	24 September 2018	

## Data Availability

Not applicable.

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
