# Peer review of "Foliar Spraying of Solanum tuberosum L. with CaCl2 and Ca(NO3)2: Interactions with Nutrients Accumulation in Tubers"

_plants, 2022, doi:10.3390/plants11131725_

Round 1
Reviewer 1 Report
Calcium is an essential macronutrient for plant growth and development. The current paper assessed the interactions of foliar fertilization with increasing concentrations of CaCl2 and Ca(NO3)2 and mineral accumulation in tubers of two potato varieties. The results are interesting. However, there are some critical issues needed to be addressed.
1. Why do the authors use 3 and 6 kg ha−1 CaCl2 or 2 and 4 kg 452 ha−1 Ca(NO3)2 solutions for foliar spraying?
2. Why do the authors only measure the phenotype under after 3rd and 4th calcium application but not all the 4 calcium applications? It is better to make it clear in certain part of the paper?
3. Why is there a significant decrease in Ca concentration in Rossi under CaCl2 3 kg ha-1 compared to other Ca application treatments?
4. Why do 4 kg ha-1 CaCl2 significantly decrease P concentration in both two cultivars compared to 2 kg ha-1 CaCl2? However, there is a different result for Na concentration between the two cultivars when applied with different CaCl2?
5. The authors need to check the statistics for Table 2 carefully. There must be some mistakes.
6. It is interesting that the Ca/Mg ratio changes under different Ca applications. Besides the Ca/Mg ratio, it is better to show the direct concentration of Ca and Mg in the epidermis and centre. Why does the Ca/Mg ratio differ between two cultivars as well as between Ca(NO3)2 and CaCl2 treatments?
Minor issues:
Line 69-72: “Besides, unbalanced nutrients interactions implicating Ca can originate complexes at the root surface or within the plant organs or compete for the site of uptake, absorption, transport, and function on plant root surfaces or within all plant tissues.” Long sentence with grammar mistakes.
Line 115: Numbers in the chemical formula must be indicated by Subscripts in Figure 1
Line 137: Numbers in the chemical formula must be indicated by Subscripts in Figure 2
Line 150: This is not Ca content but Ca concentration. It is different.
Line 181: Mineral concentrations but not Mineral content
Author Response
Reply to the reviewer
After reading and considering the reviewer perspectives of the paper “Foliar spraying of Solanum tuberosum L. with CaCl2 and Ca(NO3)2: interactions with nutrients accumulation in tubers”, the authors of the paper reply as follows:
Indications of the reviewer: ”Why do the authors use 3 and 6 kg ha−1 CaCl2 or 2 and 4 kg ha−1 Ca(NO3)2 solutions for foliar spraying?“
Reply of the authors: The authors used 3 and 6 kg ha-1 of CaCl2 or, alternatively, 2 and 4 kg ha-1 of Ca(NO3)2 considering previous studies carried out with both calcium chloride or calcium nitrate applied in Solanum tuberosum L., namely:
Seifu, Y. W., & Deneke, S. (2017). Effect of calcium chloride and calcium nitrate on potato (Solanum tuberosum L.) growth and yield. J. Hortic, 4(3), 207-211.
Hamdi, W., Helali, L., Beji, R., Zhani, K., Ouertatani, S., & Gharbi, A. (2015). Effect of levels calcium nitrate addition on potatoes fertilizer. Int. Res. J. of Eng. and Tech.(IRJET), 2(3), 2006-2013.
Additionally, the experience of our research group on this issue (Coelho, A.R.F.; Lidon, F.C.; Pessoa, C.C.; Marques, A.C.; Luís, I.C.; Caleiro, J.; Simões, M.; Kullberg, J.; Legoinha, P.; Brito, M.; Guerra, M.; Leitão, R.G.; Galhano, C.; Scotti-Campos, P.; Semedo, J.N.; Silva, M.M.; Pais, I.P.; Silva, M.J.; Rodrigues, A.P.; Pessoa, M.F.; Ramalho, J.C.; Reboredo, F.H. Can Foliar Pulverization with CaCl2 and Ca(NO3)2Trigger Ca Enrichment in Solanum tuberosum L. Tubers? Plants 2021, 10, 245. https://doi.org/10.3390/plants10020245), was further consider.
Indications of the reviewer: ”Why do the authors only measure the phenotype under after 3rd and 4th calcium application but not all the 4 calcium applications? It is better to make it clear in certain part of the paper?“
Reply of the authors: The authors only measure the leaf chlorophyll a fluorescence parameters after the last two foliar applications, to assess any physiological stresses regarding the functioning of the photosynthetic apparatus (as the harvest date approaches) triggered by the Ca treatments. In fact, depending on the extend of the potential metabolic inhibition on the photosynthetic functioning, productivity could be affected. According this hypothesis was tested.
Indications of the reviewer: ”Why is there a significant decrease in Ca concentration in Rossi under CaCl2 3 kg ha-1 compared to other Ca application treatments?“
Reply of the authors: CaCl2 application at the concentration of 3 kg.ha-1 showed a significant decrease of Ca in Rossi which revealed a different trend relatively to other treatments. In fact, we considered that at this concentration of CaCl2 eventually the synergistic and antagonist interactions with other micro and macro chemical elements could determine this trend (eventually some potential interaction with K and Mg). Besides, this is not an issue that might be easy to explain since we must also consider the metabolic differences among the potato genotypes. Nevertheless, the aims of our research were not focused on this particular aspect but rather in more general trend of nutrient accumulation in tubers.
Indications of the reviewer: ”Why do 4 kg ha-1 CaCl2 significantly decrease P concentration in both two cultivars compared to 2 kg ha-1 CaCl2? However, there is a different result for Na concentration between the two cultivars when applied with different CaCl2?“
Reply of the authors: We think the reviewer wanted to indicate CaCl2 treatments (3 and 6 kg ha-1). However, P content in both cultivars did not significantly decrease in both treatments applied with CaCl2 (please note the statistics applied for P). Relatively to the interaction between Na e CaCl2 the trend is similar when compared to the control (a decrease was found in both treatments – 3 and 6 kg.ha-1 in Agria and Rossi cultivars). Nevertheless, the different values of Na accumulation, particularly with CaCl2 with 6 kg.ha-1 treatment between both cultivars can be attributed to different genotypic characteristics.
Indications of the reviewer:” The authors need to check the statistics for Table 2 carefully. There must be some mistakes.“
Reply of the authors: The authors checked the statistic for Table 2, there was in fact an error in the uniformization of statistic in Na content in Agria variety.
Indications of the reviewer: ”It is interesting that the Ca/Mg ratio changes under different Ca applications. Besides the Ca/Mg ratio, it is better to show the direct concentration of Ca and Mg in the epidermis and centre. Why does the Ca/Mg ratio differ between two cultivars as well as between Ca(NO3)2 and CaCl2 treatments?“
Reply of the authors: Ratio Ca/Mg was used because Mg concentration is rather stable in tubers which allowed us to better compare the different accumulation trend in Ca using both fertilizers. Nevertheless, the absolute values of Mg can be attained in the legend of the Figure 4 and considering that we are showing a ratio on the contents of Ca all the values can be obtained by the readers of the paper. Besides, we decide to use that rates for comparative analysis because it has long been reported that in lot of plants Mg has an antagonistic interaction with Ca (please see Fageria, V. D. (2001). Nutrient interactions in crop plants. Journal of plant nutrition, 24(8), 1269-1290). Again, the different Ca/Mg ratios between cultivars are closely linked, according to our knowledge, with genotypes specificity (please note that the major concern of this study was to see if the accumulation of Ca prevails in the epidermis or in the centre of the tubers which we found to be similar in both cultivars).
Indications of the reviewer: ”Line 69-72: “Besides, unbalanced nutrients interactions implicating Ca can originate complexes at the root surface or within the plant organs or compete for the site of uptake, absorption, transport, and function on plant root surfaces or within all plant tissues.” Long sentence with grammar mistakes.“
Reply of the authors: The sentence of Line 69 to 72 has changed from “Besides, unbalanced nutrients interactions implicating Ca can originate complexes at the root surface or within the plant organs or compete for the site of uptake, absorption, transport, and function on plant root surfaces or within all plant tissues.” to “Besides, unbalanced nutrient interactions which implicate Ca can originate complexes. These complexes can be at the root surface, within the plant organs and can compete for the site of uptake, absorption, transport, and function in those tissues (on plant root surfaces or within all plant tissues).”
Indications of the reviewer: ”Line 115: Numbers in the chemical formula must be indicated by Subscripts in Figure 1
Line 137: Numbers in the chemical formula must be indicated by Subscripts in Figure 2“
Reply of the authors: Regarding Line 115 and Line 127 the correction was made in figures 1 and 2.
Indications of the reviewer: ”Line 150: This is not Ca content but Ca concentration. It is different.
Line 181: Mineral concentrations but not Mineral content“
Reply of the authors: Regarding the Line 150, the authors made the change of “Ca content” to “Ca concentration” and in Line 181 the authors changed to “Mineral concentrations” instead of “Mineral content”.
Reviewer 2 Report
Thank you for the manuscript.
The manuscript addresses an imminent issue of the effect of under or overfertilization on the plant performance and balance of other nutrients. This is important to prevent eutrophication issues in the environment.
However:
1) Among all the nutrient interactions, why N was not quantified. It is one of the major macronutrients.
2) Was the calcium in control fields measured. Perhaps making the soil deficient in calcium prior to this study could have resulted in better significant results in terms of photosynthesis efficiency, plant yield, and nutrient interactions.
3) The tables are numbered wrongly. In my view, the tables should be depicted as graphs and with dot plots rather than bar plots to see the data spread. The table is too complex to read and see the differences. In the second table, the data for each nutrient can be presented in an individual graph.
4) Minor grammatical errors should be addressed to make the text clear.
5) The CONCLUSION is just a compilation of the RESULTS section. Please shorten it and prepare a short summary of your outcomes in 2-3 statements plus the future outlook of this study to write the CONCLUSION.
Author Response
Reply to the reviewer
After reading and considering the reviewer perspectives of the paper “Foliar spraying of Solanum tuberosum L. with CaCl2 and Ca(NO3)2: interactions with nutrients accumulation in tubers”, the authors of the paper reply as follows:
Indications of the reviewer: ”Among all the nutrient interactions, why N was not quantified. It is one of the major macronutrients.“
Reply of the authors: Our major interest was to fully study some interactions between Ca using two fertilizers and contrasting cultivars of tubers (which we already know to occurred from our previous study). In this context, we did not measure the levels of N because this would implicate a very detail study as the organic and inorganic fractions of this nutrient (besides, the physiological implications at uptake, translocation and accumulation in tissues, would have to be integrated with the metabolic chelation in different kinds of molecules). Additionally, it has long been reported that Ca accumulation significantly differs when NO3- or NH4+ is used as fertilizer, which could become an additional problem to study this nutrient (please see Kawasaki, T. Metabolism and Physiology of Calcium and Magnesium. In Science of the Rice Plant; Matsuo, T., Kumazawa, K., Ishii, R., Ishihara, K., Hirata, H., Eds.; Food and Agricultural Policy Research Center: Tokyo,Japan, 1995; 2:412-419.)
Indications of the reviewer: ”Was the calcium in control fields measured. Perhaps making the soil deficient in calcium prior to this study could have resulted in better significant results in terms of photosynthesis efficiency, plant yield, and nutrient interactions.“
Reply of the authors: Calcium in the field was measured before of the implementation of the culture and prior any foliar application, the concentration is presented in ” Materials and Methods, 4.1. Experimental fields“, being the Ca content in Agria and Rossi fields of 0.39 and 0.71%, respectively.
Indications of the reviewer: ”The tables are numbered wrongly. In my view, the tables should be depicted as graphs and with dot plots rather than bar plots to see the data spread. The table is too complex to read and see the differences. In the second table, the data for each nutrient can be presented in an individual graph.“
Reply of the authors: The number of the tables was corrected. Concerning to the graphs we think that bar plots can furnish a better perspective that dot plots since makes the reading od the paper easier. Nevertheless, the readers of the paper can have a general perspective about the data spread since we introduced in each case the standard error associated to the means values.
Indications of the reviewer: ”Minor grammatical errors should be addressed to make the text clear.“
Reply of the authors: The authors made grammatical corrections throughout the paper.
Indications of the reviewer: ”The CONCLUSION is just a compilation of the RESULTS section. Please shorten it and prepare a short summary of your outcomes in 2-3 statements plus the future outlook of this study to write the CONCLUSION.“
Reply of the authors: We simplified in a minor extended the conclusion, yet we decided to keep most of the conclusion dealing with nutrient interactions. In fact, in a simple way this conclusion allows the readers of the paper to have and easier perspective of the study. Besides, if we delete that information the conclusion would become to much superficial according to our point of view.
Round 2
Reviewer 1 Report
All my concerns are well adressed. I have no more commends.
Reviewer 2 Report
Among my concerns, you have not addressed the dot plots issue. Anyways given the standard error is indicated, it is permissible.
I think the manuscript is okay to be accepted at this point although some of the major points were not addressed.
This manuscript is a resubmission of an earlier submission. The following is a list of the peer review reports and author responses from that submission.
Round 1
Reviewer 1 Report
The article deals with a study on the CaCl2 and Ca(NO3)2 interactions with nutrient accumulation in tubers. The aim of the study is interesting, however, the manuscript has serious flaws, so I recommend its rejection.
The different sections are not well related and discussed. You must put more effort into linking the results of the photosynthesis, the nutrient analysis, and the tuber size and color. You do not include any results related to the potato yield per treatment, even though they are important to assess the effect of the treatments. You do not include the concentration of Ca and Mg in different parts of the tuber. Moreover, the manuscript has several flaws that highlight a lack of attention that decreases its final value.
Authors should also consider the comments below, among others.
Introduction
I do not have major concerns about the introduction section except for providing more information about tubers and Ca. Moreover, if you consider in the results section that the photosynthetic performance is important in your experiment, why you do not include any reference in the introduction section.
In my opinion, the introduction is ok as it is and authors should consider if they want to maintain the sections related with the photosynthetic performance as they are not able to relate the values with the nutritional profile of the potatoes, they do not show values related with the sugars or starch concentrations/content in them.
It is the same case for the color measurements.
Materials and Methods
L90: Plant density?
L99: Adequate irrigation conditions. Can you describe them? Two plots were considered so, did they follow the same irrigation practices?
Calcium is translocated via the xylem, so its distribution is highly influenced by the plant water status.
L102: It makes no sense to talk about the daily average rainfall, however, it is important to know when did it rain, did it rain right after any spraying?
L110: It is not clearly explained the number of treatments (5) and replicates assayed, and the number of plants per replicate. It would be useful to include a table with the treatment the number of foliar sprayings of each treatment, the product, the concentration, the date and the number of plants.
A map with the treatment distribution within your plots would also be helpful.
L136: Define "serie", how many series are there? It will help to understand when you write "in three replicates of three independent series".
L150: was then determined?
L160: how many slices?
L173: Height and diameter were measured in ... how many potatoes per plant? how many plants per replicate? how many replicates per treatment?
Results
Figure 1: There were no significant differences in any of the leaf chlorophyll a parameters considered between treatments on Aug, 2nd (Third spray). Why did you decide to show the values measured on the third and fourth application dates but not the first and second applications? It would be better a line graph with the evolution of each treatment.
Figure 2: Same as figure 1. There are no significant differences in Yno neither between treatments nor between days. Why do you want
I don't see the point of cutting the Y-axis at the values that you did. It does not help to visualize the results.
Figure 3: Are there differences in Ca content between Agria and Rossi?
What about Ca concentration? Are there differences among treatments if you consider the Ca concentration?
L268: You must show the content of Ca and Mg in every section. It makes no sense to discuss the values of the ratio Ca/Mg when you do not want to show the values.
Figure 4: What about the CTL treatment? Are there differences between 'Agria' and 'Rossi' for the CTL treatment?
Why only two sections for Ca if it is the main mineral evaluated in the paper?
L219-354. Every statement of the first paragraph has an exception! It is hard to read and you can’t identify the main results.
You should rewrite the information related to table 2, try to be more concise, you do not have to repeat all the values that you already provide in the table. It is an important part of the results section that deserves a better description.
Table 2. Minerals? Carbon, Oxygen and Hydrogen!!!
You must include the yield. Differences in yield will have a huge impact in the size and nutrient content.
L386. According to Table 3, both CaCl2 concentrations were significantly higher than Ctl and Ca(NO3) treatments for 'Rossi' potatoes.
L396. Discussion?
L406. Did you measure plant gas exchange?
L415. Tuber yield is an important part of your results.
L416. Are there any papers that used higher concentrations than those that you used?
L438. Ca accumulation in Ca(NO3)2 'Rossi' tubers did not significantly increase K contents according to Table 1.
Please, consider that when you are discussing K and Ca relationship.
Where is the discussion of the differences in the Ca/Mg ratio?
What about the a* parameter for 'Rossi'- CaCl2
L487. This is not true according to your data. Please check table 3. 'Rossi' CaCl2 potatoes had higher values in the epidermis than in the center, section 2 vs section 1 for both concentrations.
L514. You do not show the values of Ca in the epidermis and the center. You only show the ratio between Ca/Mg. You state that the lowest values of Ca were found in the epidermis but you did not show any values in the result section. Moreover, it seems that there is not a solid pattern regarding the concentration of Zn in the different sections of the tuber. While Ctl tubers showed the lowest Zn concentrations in the epidermis, other treatments showed the highest concentrations of Zn in the epidermis.
L531. This is not true according to your data. Please, check table 2. 'Rossi' color resulted affected by the CaCl2 concentrations.
Conclusions
L536. "increased Ca accumulation in tubers (mostly in the parenchyma tissues located in the center of the equatorial region)" Where can I see these results? You do not provide any result of the Ca or Mg concentration in different sections of the tuber.
Reviewer 2 Report
Coelho et al. discussed the effect of Ca foliar fertilization on the distribution of the minerals and on the photosynthesis of potato plants. The article while appropriate for Plants, should undergo a substantial revision in order to be considered for publication. Please find my comments below.
Line 62-69 please revise for improved consistency, the paragraph is rather vague. Specify the interactions and competitions between Ca and other nutrients.
Line 71 revise “negative interactions”
Line 72 revise “bypass the use of soil….”
Line 74 elaborate on the effect of Ca on tuberization
Line 112 explain “test system”
Line 117 provide the number of plants
Please improve the quality of the figures, especially the depiction of the statistical analysis
The photosynthetic parameters of the cv Rossi are missing, this can be confusing and should be resolved either by including the data or by introducing a plausible explanation for the missing data.
Introduce more data in the legend of figure 4 and provide all the required information for the reader to understand the findings.
Line 396 revise the title
Line 399 revise for improved consistency
Lines 412-413 please clarify
Line 415 please provide the referenced data
Line 421 please revise
Line 426 specify “non toxic levels”
Lines 439-440 revise as non such data is provided
The effect of the treatments on the quality of the tubers is not discussed in the discussion part.